# Optimization Design and Control of Six-Phase Switched Reluctance Motor with Decoupling Winding Connections

**Wei Qiao** [1], **Shouyi Han** [1], **Kaikai Diao** [2,*] and **Xiaodong Sun** [2,*]

1. School of Electrical and Information Engineering, Jiangsu University, Zhenjiang 212013, China
2. Automotive Engineering Research Institute, Jiangsu University, Zhenjiang 212013, China
* Correspondence: diaokaikai@ieee.org (K.D.); xdsun@ujs.edu.cn (X.S.)

**Abstract:** In this paper, a design optimization method is proposed to reduce the current asymmetric and consequent torque ripple for a six-phase Switched Reluctance Motor (SRM). First, the inconsistent current phenomenon of the investigated SRM is introduced, and the relationship between the magnetic distribution and the phase currents is investigated by magnetic circuit analysis. Then, for the reduction in computational cost, a surrogate model is utilized to establish the response surface model between the optimization objectives and variables. Furthermore, multiobjective optimization is performed based on structural design optimization and asymmetric control, and the best design solution is selected for the drive system. Compared with the traditional symmetric control, it can be found that the proposed asymmetric control can mitigate the inconsistent phase currents and reduce the torque ripple. Finally, a prototype motor is manufactured and tested. Both the simulation and experimental results verify the effectiveness and the reasonability of the analysis and the optimization.

**Keywords:** finite-element model (FEM); multiobjective optimization; switched reluctance motor

## 1. Introduction

Switched Reluctance Motors (SRMs) have become promising candidates and aroused considerable attention due to their inherent simple structure, high reliability, and the absence of permanent magnets [1–3]. A four-quadrant operation can be easily realized for SRM through angle adjustment. Thus, it exhibits great application potential in the field of vehicles [4–6]; however, due to the doubly salient structure, the torque ripple and vibration noise of SRMs are much larger than other types of motors, which limits their promotion in some specific fields; therefore, in recent decades, suppression of torque ripple and vibration noise has become one of the main topics of SRMs [7–9].

Scholars have carried out a lot of work on the suppression of torque ripple, mainly including two main aspects, i.e., the design optimization of the structure [10–12] and the development of control strategy [13–15]. In the previous works, it could be found that increasing the number of phases of SRMs can significantly reduce the torque ripple and vibration noise [16,17]. Compared with the traditional three-phase SRM, the torque ripple of the six-phase SRM can be reduced by two-thirds [18,19]. Since the electromagnetic torque of SRMs is mainly generated by the electromagnetic pulling force, the distribution of the magnetic field has a great influence on motor performances [20].

Furthermore, it was proved that the distribution of the magnetic field is closely related to the winding connections [19,21]. In [22], the concept of forward series and reverse series of the SRM windings is proposed. The self-inductance, mutual inductance, and static torque of the three-phase SRM with two types of windings in positive series and reversed series are analyzed and compared. Meanwhile, the different magnetic pole distributions of the four-phase 8/6 structure SRM were investigated in [23,24]. It was proved that when the alternating arrangement of NS magnetic poles was adopted in the odd-phase SRM, the phases were decoupled [23]. The different excitation modes of the four-phase 8/6



stator/rotor poles SRM under the two magnetic pole distributions are investigated, and a new method was proposed to reduce torque ripple by setting different current chopping limits [24]. In [21], possible winding arrangements for six-phase SRMs are presented and analyzed. Moreover, in [25], the combination of magnetic poles under different conduction modes was proposed, and the influence caused by the winding connections on static torque was further investigated. Through the comprehensive comparison of different winding connections, it has been found that the NNSS alternating winding arrangement exhibit the best overall performance; however, at the same time, asymmetry current phenomenon occurs in the six-phase SRM [21,26].

In summary, there is a coupling phenomenon between the phases when the winding connection method with reverse series was applied in the even-numbered phase SRM, which reduces the fault-tolerant ability of SRMs. Moreover, the magnetic field distribution of the system will be asymmetric if the decoupling winding connection is utilized, leading to the asymmetry of the phase currents in the even-numbered phase SRM. This will further increase the torque ripple of the system [27,28].

In this paper, a novel multiobjective optimization method incorporated with an asymmetric control is proposed to reduce the inconsistent phase currents and consequent torque ripple for a six-phase 12/10 SRM. The main contributions of this paper are listed as follows.

(1) The asymmetric current phenomenon for the investigated SRM is introduced and explained by the magnetic circuit analysis.
(2) A novel system-level multiobjective optimization method is proposed by considering the structural variables and the turn-on angles and conduction width in the proposed asymmetric control.

The remainder of this paper is organized as follows. In Section 2, the current asymmetry caused by the forward series winding connection is investigated and its influence on torque ripple is analyzed. Section 3 describes the applied surrogated modeling method. Then, in Section 4, a system-level multiobjective optimization is proposed to reduce the current asymmetry and torque ripple of the drive system. Results and experimental verification are presented in Section 5, followed by the conclusion in Section 6.

## 2. The Phenomenon of Current Asymmetry

### 2.1. Asymmetric Current Phenomenon

In the traditional SRMs with an odd number of phases, such as the three-phase 6/4 and the five-phase 10/8 SRMs, the magnetic poles are alternately arranged at NS when the windings are connected in the forward direction. The polarities between two stator teeth of one phase are opposite; thus, a flux loop is generated between the two stator teeth, the interphase coupling can be ignored, and the magnetic field distribution is symmetric.

For the SRMs with an even number of phases, a six-phase 12/10 SRM is taken as an investigated example to illustrate the asymmetric current phenomenon, as shown in Figure 1. The relevant main parameters of the motor are presented in Table 1. When the winding arrangement is the NS alternative shown in Figure 1a, the polarities between the two stator teeth of each phase are the same. For example, the polarities of the teeth $A_1$ and $A_2$ are both N polar; therefore, flux loops with the adjacent phases will be generated when the motor is excited, resulting in a large coupling between the adjacent phases. Meanwhile, the magnetic field distributions are the same when two adjacent phases are simultaneously excited, such as phases A and B or phases B and C, as shown in Figure 1a. In addition, when the decoupling winding arrangement is applied, such as the NNSS alternative shown in Figure 1b, the polarities of the two stator poles in each phase are opposite. Under this situation, if two adjacent phases are excited simultaneously, flux loops with different distances will be generated, as presented in Figure 1b. For instance, a long flux path is formed when phases A and B are simultaneously excited, as shown by the red dotted line in Figure 1b. While a quite shorter one is formed when phases B and C are simultaneously excited, as shown by the green dotted line in Figure 1b; therefore, the

magnetic field distribution is asymmetric when the NNSS alternative winding arrangement is adopted for the six-phase SRM.

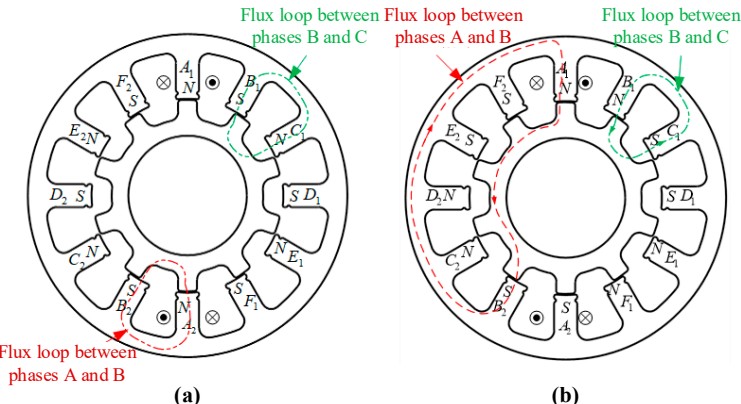

**Figure 1.** Magnetic poles and flux distribution in a six-phase SRM under winding connections with (**a**) NS alternative and (**b**) NNSS alternative.

**Table 1.** Main parameters of six-phase SRM.

| Parameters | Value | Parameters | Value |
|---|---|---|---|
| Stator poles | 12 | Stator tooth width ($b_{ps}$/mm) | 5.6 |
| Rotor poles | 10 | Rated speed (r/min) | 3000 |
| Stator outer diameter ($D_s$/mm) | 80 | Rated voltage/V | 24 |
| Rotor outer diameter ($D_a$/mm) | 47.4 | Stack length ($l_a$/mm) | 70 |

Figure 2 shows the current and torque waveforms achieved by the finite-element models (FEMs) when the investigated six-phase SRM with NNSS alternative connection. It can be found that the current waveforms of phases A, C, and E are similar, and so are the other three phases. The maximum current values in phases A, C, and E are greater than the remaining three phases. It should be noted that, as presented in Figure 2, torques at points 1 and 3 are the maximum torques corresponding to the maximum current positions, and torques at points 2 and 4 are the minimum values at the intersection of the phase currents. Since the current value in phase A is generally greater than that with the same position of phase B, as a result, the torques at points 1 and 2 are larger than those at points 3 and 4, respectively; therefore, it can be concluded that the asymmetry of the phase currents aggravates the torque ripple of the system.

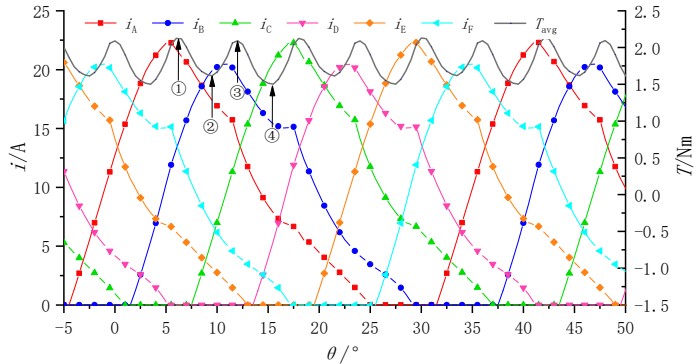

**Figure 2.** Current and torque waveforms of the investigated six-phase SRM under NNSS alternative winding connection.

### 2.2. Magnetic Circuit Analysis

The stator yokes between phases A and B are selected as the analysis objects. The polarities of adjacent poles between phases A and B could be the same as NN or the opposite NS, which correspond to the magnetic field distributions between phases A and B, and phases B and C in Figure 1b, respectively. Assuming that the magnetic flux leakage is ignored, the magnetic circuits under two kinds of magnetic pole arrangements are presented in Figure 3.

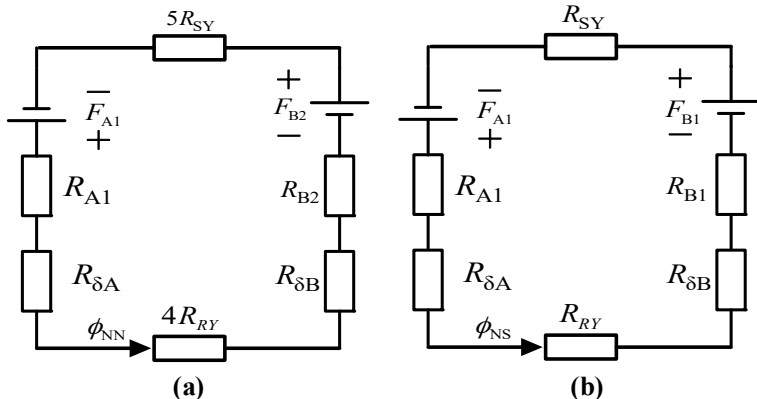

**Figure 3.** Magnetic circuit of six-phase SRM: (**a**) NN and (**b**) NS.

In Figure 3, $R_{SY}$ and $R_{RY}$ are the reluctances of stator yoke and rotor yoke, respectively, $R_{A1}$, $R_{B1}$, and $R_{B2}$ are the sum of the reluctances stator teeth and rotor teeth corresponding to poles $A_1$, $B_1$, and $B_2$ and their aligned rotor poles, respectively, $R_{\delta A}$ and $R_{\delta B}$ are the air gap reluctances of phase A and phase B, respectively, $F_{A1}$, $F_{B1}$, and $F_{B2}$ are the magnetomotive force excited by the windings on poles $A_1$, $B_1$, and $B_2$, respectively, and $\phi$ is the magnetic flux.

Since SRM is the type of symmetric motor and each stator pole is excited with the same current condition, the magnetomotive forces shown in the two figures in Figure 3 are equal, which leads to

$$\frac{\phi_{NN}}{\phi_{NS}} = \frac{R_{A1} + R_{B1} + R_{\delta A} + R_{\delta B} + R_{SY} + R_{RY}}{R_{A1} + R_{B2} + R_{\delta A} + R_{\delta B} + 5R_{SY} + 4R_{RY}} \tag{1}$$

Assume that the magnetic flux density of the stator yoke is B = 1.5 T, and the magnetic permeability of the silicon steel sheet is 1500 times that of the vacuum. Based on the parameters in Table 1. The relationships between the other reluctances with the airgap reluctance $R_{\delta A}$ can be represented by

$$R_{\delta A} = R_{\delta B}/2 = 115R_{A1} = 15R_{SY} = 22R_{RY} \tag{2}$$

Furthermore, based on (1) and (2), it can be deduced that

$$\phi_{NN}/\phi_{NS} \approx 0.88 \tag{3}$$

From Equation (3), it can be found that under the same current conditions, the magnetic flux of the stator yoke between NN is smaller than that between NS. It means that the machine with NS arrangement is easier to be saturated than that with NN arrangement. During the excitation mode, the current of phase A can be calculated by

$$i_A = (U_S - u_L)/R = \left(U_S - \frac{N_{ph}d\phi_A}{dt}\right)/R \tag{4}$$

where $i_A$, $U_s$, $u_L$, $R$, $N_{ph}$, and $\phi_A$ represent the current of phase A, DC voltage, phase winding voltage, resistance, number of turns, and flux linkage of phase A, respectively.

It is acknowledged that with the increase in the magnetic flux density, the change rate of the magnetic field gradually decreases. From Formulas (1)~(4), the excitation current rises faster with NS arrangement than that with NN arrangement, which leads to the asymmetric phase currents in six-phase SRM with NNSS winding connection.

Based on the above analysis, it can be concluded that when the windings are connected in forward series with the magnetic pole distribution alternately arranged by NNSS, the excitation currents are unequal. In other words, the current asymmetry is caused by the asymmetric magnetic flux density of the yokes; therefore, the asymmetric phase currents can be relieved by optimizing the stator yoke width or selecting a suitable control scheme.

Set the total magnetic resistance except for the stator yoke as $\Sigma R$ in Figure 3b, and then the magnetic flux in the circuit can be expressed as

$$\begin{cases} \phi_{\mathrm{NS}} = BS_{\mathrm{SY}} = \dfrac{N_{\mathrm{ph}}i_{\mathrm{A}} + N_{\mathrm{ph}}i_{\mathrm{B}}}{R_{\mathrm{SY}} + \Sigma R} \\ R_{\mathrm{SY}} = \dfrac{l_{\mathrm{a}}}{\mu_{\mathrm{Fe}}S_{\mathrm{SY}}} \end{cases} \tag{5}$$

where $B$, $S_{\mathrm{sy}}$, $N_{\mathrm{ph}}$, $i_{\mathrm{B}}$, $l_{\mathrm{a}}$, and $\mu_{\mathrm{Fe}}$ represent magnetic flux density, the cross-sectional area of stator yoke, current of phase B, lamination length, and permeability of iron, respectively.

Equation (5) can be rewritten as

$$B = \frac{N_{\mathrm{ph}}i_{\mathrm{A}} + N_{\mathrm{ph}}i_{\mathrm{B}}}{\left( \dfrac{l_{\mathrm{a}}}{\mu_{\mathrm{Fe}}} + S_{\mathrm{SY}}\Sigma R \right)} \tag{6}$$

Equation (6) shows that under the same number of turns and stack length, the magnetic flux density can be reduced in two ways, i.e., reduction in the magnetomotive force and increase in the stator yoke area. The first way can be realized by shortening the overlapping angle of conduction between adjacent phases or reducing the conduction width of each phase. On the other way, it can be realized by appropriately shortening the stator yoke width, which can be represented by the ratio of the yoke width to the tooth width; therefore, in this investigated example, the turn-on angle $\alpha_{\mathrm{on}}$, the turn-off angle $\alpha_{\mathrm{off,}}$ and the yoke width ratio $\lambda_{\mathrm{hcs}}$ are selected as the optimization parameters.

## 3. Surrogate Modeling Method

### 3.1. Sample Data Collection

The size definition of the prototype is shown in Figure 4, in which the pole arc coefficients of the stator and the rotor, and the inner and outer diameters of the stator and rotor are fixed values. Define the yoke width ratio $\lambda_{\mathrm{hcs}} = h_{\mathrm{cs}}/b_{\mathrm{ps}}$, the stator slot area $S$ can be expressed as

$$S = \frac{\pi}{48}\left( (D_{\mathrm{S}} - 2b_{\mathrm{ps}}h_{\mathrm{cs}})^2 - D_{\mathrm{si}}^2 \right) - \lambda_{\mathrm{hcs}}b_{\mathrm{ps}}^2 \tag{7}$$

Once the number of turns of each phase is fixed, the resistance of each phase changes with the width of the stator yoke under the condition of the same slot fill factor. Thus, the resistance of each phase is set as dynamic resistance, which can be represented by

$$r_{\mathrm{sp}} = \frac{4N_{\mathrm{ph}}^2 \rho \left( l_{\mathrm{a}} + b_{\mathrm{ps}} \right)}{k_{\mathrm{u}}S} \tag{8}$$

where $\rho$ is the resistivity and $k_{\mathrm{u}}$ is the slot fill factor.

It can be concluded from (7) and (8) that the resistance of each phase is related to the optimized parameter $\lambda_{\mathrm{hcs}}$. The relevant parameters of the motor are shown in Table 1, and the optimization variables are shown in Table 2. It should be noted that the angle position control (APC) is taken as the control method. Thus, the turn-on angle and the conduction width are the control parameters for further optimization. In this paper, the simulation $0°$ is defined as the position where the stator teeth of phase A is aligned with the centerline of the rotor teeth (the y-axis position shown in Figure 4).

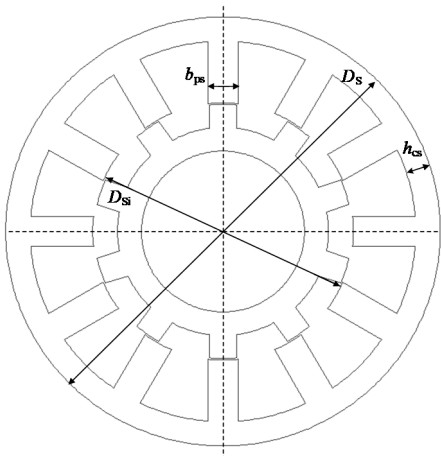

**Figure 4.** Schematic of parameters in six-phase SRM.

**Table 2.** Optimization parameters and their range.

| Parameters | Symbol | Initial Value | Scope |
|---|---|---|---|
| Yoke width ratio ($\lambda_{\text{hcs}}$) | $x_1$ | 0.8 | [0.8, 1] |
| A-phase turn-on angle ($\alpha_{\text{Aon}}/°$) | $x_2$ | 13 | [13, 14] |
| B-phase turn-on angle ($\alpha_{\text{Bon}}/°$) | $x_3$ | 19 | [18, 19] |
| A-phase conduction width ($P_{\text{dA}}/°$) | $x_4$ | 16 | [15, 16] |
| B-phase conduction width ($P_{\text{dB}}/°$) | $x_5$ | 16 | [15, 16] |

It could be found in our recent work [26] that $\lambda_{\text{hcs}}$ exhibits a great influence on the saturation degree and too small values will cause the current distortion. Moreover, the slot fill factor will be reduced with the enlargement of $\lambda_{\text{hcs}}$; thus, the range of $\lambda_{\text{hcs}}$ is determined as [0.8, 1]. For the turn-on angle of phase A, as the aligned position is defined as 0°, the unaligned position is 18°, where 18° is half of the rotation period. For SRMs, it is acknowledged that the turn-on angle should set in advance before the unaligned position to possibly generate more power. Based on our design experience, the turn-on angle could be set around 50 electrical degrees (corresponding to 5°) ahead of the unaligned position, and the conduction angle could be 160 electrical degrees (corresponding to 16°) to avoid the negative torque. Thus, the range of the turn-on angle of phase A and its conduction width could be set around 13° and 16°, respectively. Since the phases of SRMs are excited sequentially, the turn-on angle range of phase B is added 6° (rotation period divided by the phase number) to the basis of phase A.

According to the previous analysis, the saturation of the stator yoke leads to the asymmetry of the phase currents, which intensifies the torque ripple of the system; therefore, this paper selects the average torque $T$ (Nm), the torque ripple $T_{\text{rip}}$, and the difference between the RMS value of the two adjacent phase current (represented by $I^*$) are set as the optimization objectives, where $I^*$ is generally expressed as

$$I^* = \left| \frac{I_{\text{Brms}} - I_{\text{Crms}}}{I_{\text{Brms}}} \right| \cdot 100\% \tag{9}$$

where $I_{\text{Brms}}$ and $I_{\text{Crms}}$ represent the RMS values of currents of phases B and C, respectively.

### 3.2. Kriging Model

The ranges of optimization variables are listed in Table 2. It requires a large number of samples if directly applying FEMs to reflect the relationship between optimization objectives and variables. Kriging model, one of the effective surrogate modeling methods, can be used to represent the relationship between the optimization parameters and objectives.

The relationship between the variables and the outputs can be obtained by the Kriging model based on the existing data achieved from the FEM.

Given n sample points $[x_1, x_2, \ldots, x_n]$ and their responses $[y(x_1), y(x_2), \ldots, y(x_n)]$, for a set of input parameters $x$, the response of the Kriging model can be expressed as [29–31]

$$y(x) = f(x)^T \beta + z(x) \tag{10}$$

where $f(x)^T \beta$ is the regression model, and $z(x)$ is a random error term used for the modeling of local deviation, which is usually assumed to be a vector with a mean of zero, covariance $\sigma^2$, and covariance matrix $\text{cov}_{ij}$ as

$$\text{cov}_{ij} = \sigma^2 \mathbf{R}[R(\mathbf{x_i}, \mathbf{x_j})] \tag{11}$$

where $\mathbf{R}$ is a correlation matrix that can be represented by the Gaussian correlation function matrix, and $R$ is the user-specified correlation function

$$\mathbf{R} = \begin{bmatrix} r(\mathbf{x_1}, \mathbf{x_1}) & r(\mathbf{x_1}, \mathbf{x_2}) & \cdots & r(\mathbf{x_1}, \mathbf{x_n}) \\ r(\mathbf{x_2}, \mathbf{x_1}) & r(\mathbf{x_2}, \mathbf{x_2}) & \cdots & r(\mathbf{x_2}, \mathbf{x_n}) \\ \vdots & \vdots & \ddots & \vdots \\ r(\mathbf{x_1}, \mathbf{x_1}) & r(\mathbf{x_1}, \mathbf{x_1}) & \cdots & r(\mathbf{x_1}, \mathbf{x_1}) \end{bmatrix} \tag{12}$$

where $r(x_i, x_j) = \exp\left\{ -\sum_{k=1}^{D} \alpha_k |x_{ik} - x_{jk}|^2 \right\}$, $D$ is the dimension of the design values, and $\alpha_k$ is a flexible parameter that varies randomly.

The outcomes $y(\mathbf{x})$ and $\beta$ can be estimated by using the best linear unbiased estimator in statistics

$$\begin{cases} \hat{y}(x) = f(x)^T \hat{\beta} + g(x)^T R^{-1}(y - F\hat{\beta}) \\ \hat{\beta} = (F^T R^{-1} F)^{-1} F^T R^{-1} y \end{cases} \tag{13}$$

where $F$, $r(\mathbf{x})$, and $y$ can be represented by

$$\begin{cases} F = \begin{bmatrix} f_1(x_1) & f_2(x_1) & \cdots & f_q(x_1) \\ \vdots & \vdots & & \vdots \\ f_1(x_n) & f_2(x_n) & \cdots & f_q(x_n) \end{bmatrix} \\ g(x) = \begin{bmatrix} R(x, x_1) \\ \vdots \\ R(x, x_n) \end{bmatrix} \\ y = [y(x_1), y(x_2), \ldots, y(x_n)]^T \end{cases} \tag{14}$$

By using maximum likelihood estimation, $\sigma^2$ can be estimated by:

$$\hat{\sigma}^2 = \frac{1}{n} (y - F\hat{\beta})^T R^{-1} (y - F\hat{\beta}) \tag{15}$$

Through the above analysis, the Kriging model of the 12/10 SRM can be established based on a certain number of sample points. The relationships between optimization objectives and parameters generated by the Kriging model are presented in Figure 5.

It can be found in Figure 5 that $T$ increases significantly with the advance of the turn-on angle and the increase in the conduction angle. The minimum value of $T_{\text{rip}}$ occurs at the minimum values of the conduction angles. Moreover, $T_{\text{rip}}$ and $I^*$ do not exhibit a single linear trend relationship between the conduction angles.

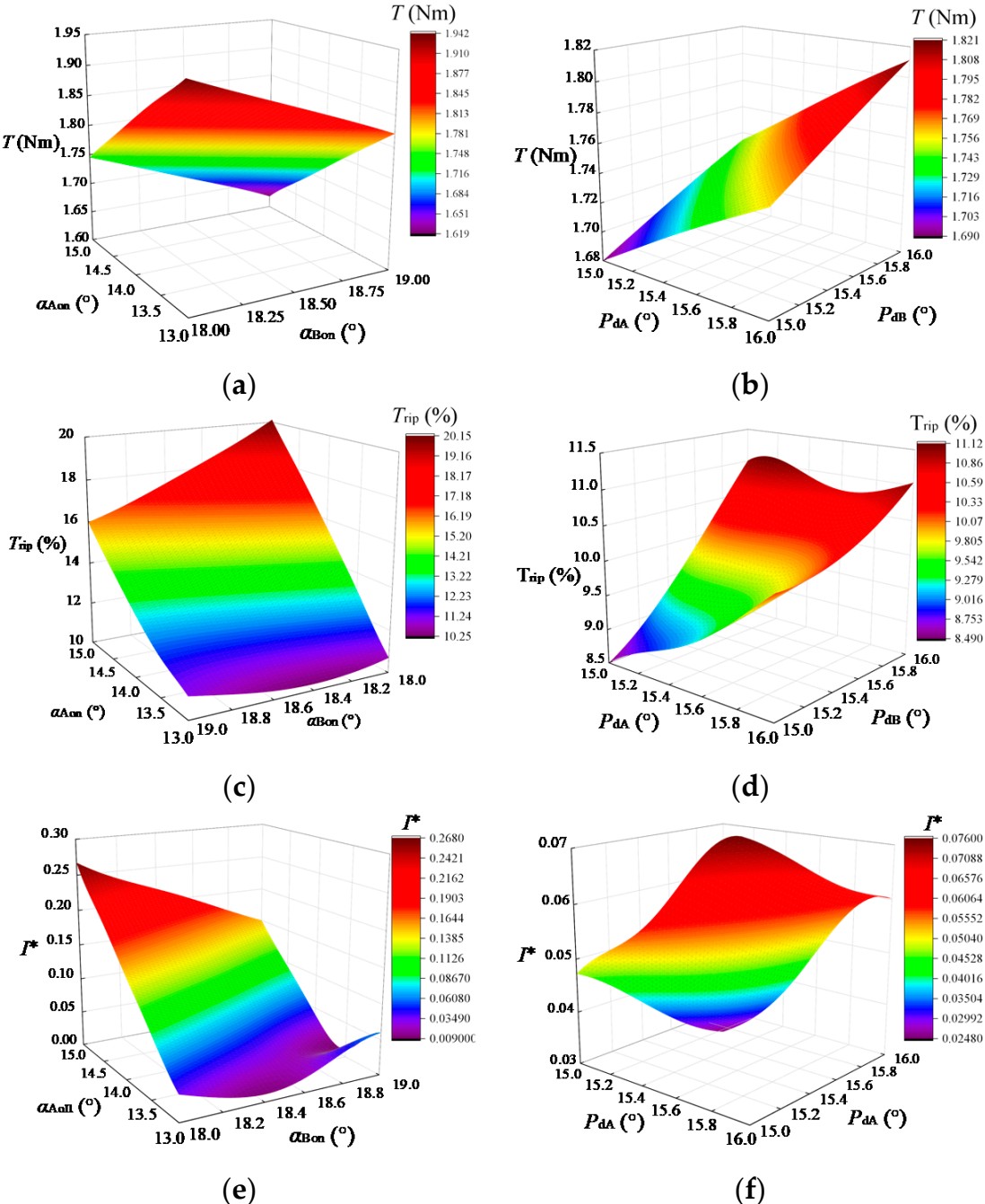

**Figure 5.** Relationships between (**a**) $T$, $\alpha_{\text{Aon}}$, and $\alpha_{\text{Bon}}$, (**b**) $T$, $P_{\text{dA}}$, and $P_{\text{dB}}$, (**c**) $T_{rip}$, $\alpha_{\text{Aon}}$, and $\alpha_{\text{Bon}}$, (**d**) $T_{rip}$, $P_{\text{dA}}$, and $P_{\text{dB}}$, (**e**) $I^*$, $\alpha_{\text{Aon}}$, and $\alpha_{\text{Bon}}$, and (**f**) $I^*$, $P_{\text{dA}}$, and $P_{\text{dB}}$.

The comparison between the Kriging model and the FEM is presented in Figure 6. A certain number of design samples are randomly selected. It can be found that the errors between the Kriging model and FEM are acceptable for all three optimization objectives. These errors are explained by two aspects: one is the inherent error when estimating the parameters such as $\beta$ and $\sigma^2$, and the other is the number of samples from the FEMs. Too few samples will cause a large deviation between the estimated and actual values of these estimated parameters in the Kriging model. Moreover, due to the nonlinear characteristics of SRMs, errors between the estimated and actual values of the objectives may be positive or negative.

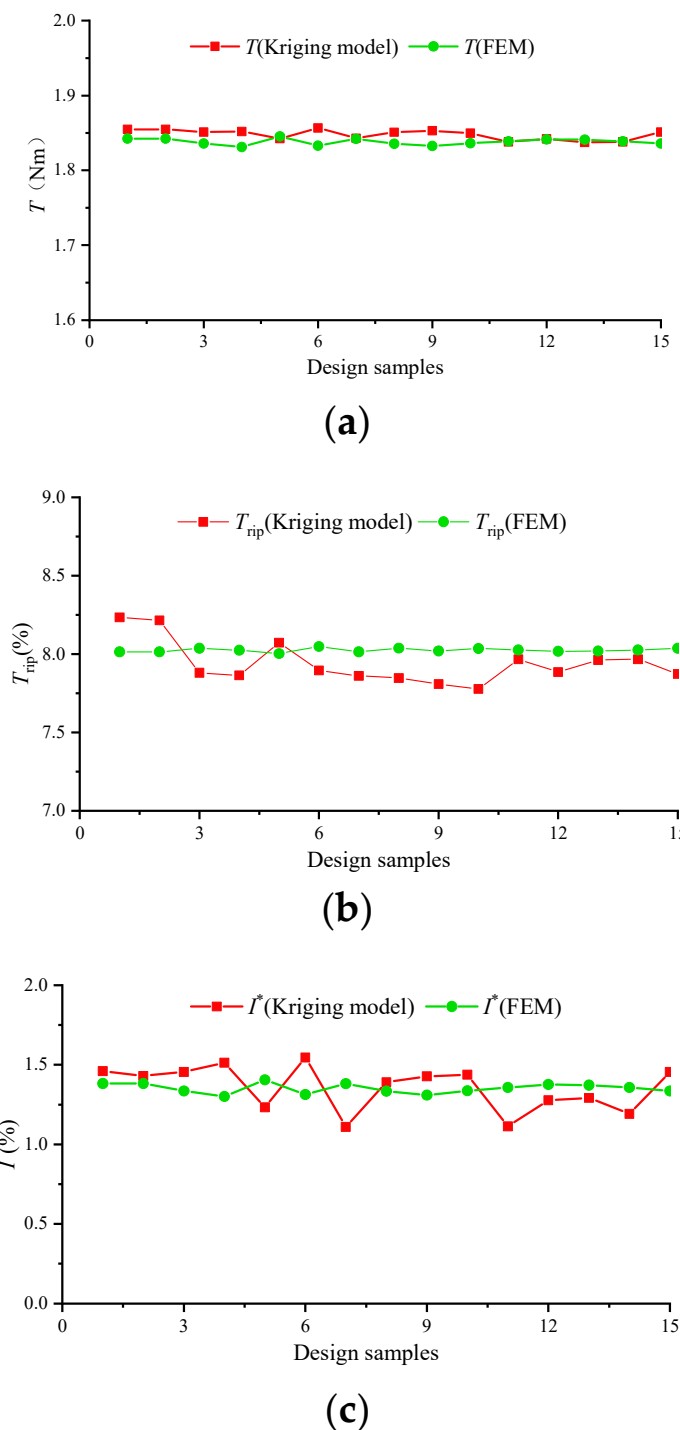

**Figure 6.** Comparison between the Kriging model and FEM of the three objectives. (**a**) $T$, (**b**) $T_{\text{rip}}$, and (**c**) $I^*$.

If the relationship between the objectives and the variables is directly established by FEM without surrogate models, it may require $10^5$ FEMs if the sampling numbers of each variable are set as 10. With the help of the surrogate model, the sampling numbers could be greatly reduced; for instance, in this example, only 1125 ($5 \times 5 \times 5 \times 3 \times 3$, where the numbers represent the sampling numbers of $\lambda_{\text{hcs}}$, $\alpha_{\text{Aon}}$, $\alpha_{\text{Bon}}$, $P_{\text{dA}}$, and $P_{\text{dB}}$, respectively. It requires only about 1% samples compared with the method of directly using the FEMs. The computational time of each FEM established in ANSYS/Maxwell is about 5 min, and the computer can run six samples simultaneously. Moreover, the running time of the Kriging

model is only a few seconds, which can be ignored compared with that of the FEMs. Thus, the total time is about 39 h; therefore, the Kriging model combined with the FEMs can be employed as the alternative for directly using FEMs to reduce the computational burden. After the establishment of the Kriging model, the relationship between the objectives and the parameters can be expressed by it, which provides the foundation for the further optimization process illustrated in Section 4.

## 4. Multiobjective Optimization

It can be found in Figure 5 that there is a conflict in the solution set between increasing the output torque and reducing the torque ripple; therefore, the optimization algorithm should be utilized to find a multiobjective optimal result based on the data predicted by the Kriging model. NSGA-II is an improved non-dominated sorting genetic algorithm, which has been widely utilized in multiobjective optimization problems [32–34]. It was first proposed by Srinivas in 1994 [35]. The procedures of NSGA-II mainly consist of non-dominated sorting and genetic operator operation. The detailed procedures are presented as follows.

(1) Initialization of the population. In this step, the population and the offspring population are created randomly.
(2) Classification of the population into nondominated levels. The nondominated number $n_\text{p}$ of each individual can be figured out, and $n_\text{p} = 0$ if the individual is in the first nondominated level. The rest individuals are repeated to complete the classification of the whole population.
(3) Calculation of crowding distance. Each Pareto point is ranked according to its objective $f_\text{m}$. The crowding distance $n_\text{d}$ of the first and last Pareto points are set as infinite, and the crowding distance of the rest Pareto points can be calculated as

$$n_{\text{d}(i+1)} = n_{\text{d}(i)} + (f_\text{m}(i+1) - f_\text{m}(i-1))/(f_\text{m}^{max} - f_\text{m}^{min})$$ (16)

(4) Creation of offspring population. The parent population is created by the nondominated levels and crowding distance, and the offspring population is generated through selection, crossover, and mutation.
(5) Repetition of steps (2)–(4). The step (2)–(4) are repeated until they reach the appointed generation and the Pareto front can be achieved.

The output torque, torque ripple, and the normalization value of phase current are defined as optimal objectives, and the current density is defined as a constraint. The multiobjective model can be defined as

$$\begin{cases} \text{Function}: & \min\{-T(Z), T_\text{rip}(Z), I^*(Z)\} \\ \text{Constraint}: & J_\text{A} \leq 10, \ J_\text{B} \leq 10 \\ s.t. & Z = \{x_1, x_2, x_3, x_4, x_5\} \end{cases}$$ (17)

where $J_\text{A}$ and $J_\text{B}$ are phase current densities of phases A and B, respectively.

After using the multiobjective optimization algorithm, the Pareto front [36,37] can be obtained, as presented in Figure 7. It can be found that the Pareto front points are scattered when the population size is defined as 200, while it is more concentrated when the population size is 10.

The optimal solutions selected from the Pareto front are listed in Table 3 when the number of individuals is defined as 10, and each design satisfies the constraint for current density. The comparison between the optimization results and FEM is presented in Table 4, in which $T\_\text{S}$, $Trip\_\text{S}$, and $I^*\_\text{S}$ represent the results from the FEM. It can be found the errors between these are acceptable.

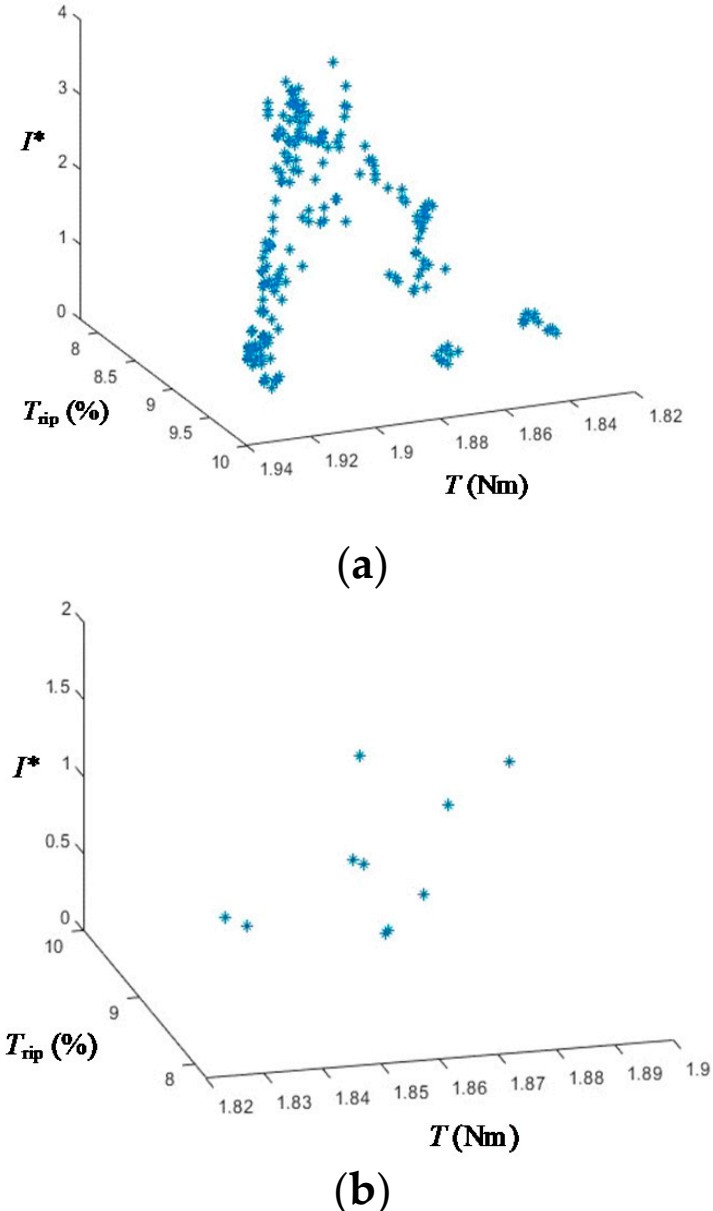

**Figure 7.** Pareto front when population sizes are (**a**) 200 and (**b**) 10, respectively.

**Table 3.** Optimal results of Pareto points.

|  | $\lambda_{hcs}$ | $\alpha_{Aon}$ | $\alpha_{Bon}$ | $P_{dA}$ | $P_{dB}$ |
|---|---|---|---|---|---|
| Design 1 | 0.909 30 | 13.002 | 18.543 | 15.013 | 15.113 |
| Design 2 | 0.895 98 | 13.000 | 18.564 | 15.027 | 15.065 |
| Design 3 | 0.915 14 | 13.000 | 18.599 | 15.000 | 15.159 |
| Design 4 | 0.893 55 | 13.036 | 18.549 | 15.820 | 15.327 |
| Design 5 | 0.911 77 | 13.015 | 18.704 | 15.216 | 15.274 |
| Design 6 | 0.890 19 | 13.037 | 18.545 | 15.667 | 15.303 |
| Design 7 | 0.906 14 | 13.001 | 18.523 | 15.024 | 15.049 |
| Design 8 | 0.91 508 | 13.000 | 18.601 | 15.028 | 15.166 |
| Design 9 | 0.895 79 | 13.000 | 18.562 | 15.084 | 15.070 |
| Design 10 | 0.909 39 | 13.000 | 18.539 | 15.247 | 15.024 |

**Table 4.** Comparison between the optimization results and FEM.

| | $T$ | $T\_S$ | $T_{rip}$ | $T_{rip\_S}$ | $I^*$ | $I^*\_S$ |
|---|---|---|---|---|---|---|
| Design 1 | 1.843 | 1.842 | 7.860 | 8.014 | 1.210 | 1.381 |
| Design 2 | 1.825 | 1.826 | 7.958 | 8.016 | 1.152 | 1.266 |
| Design 3 | 1.845 | 1.845 | 7.789 | 7.981 | 1.340 | 1.403 |
| Design 4 | 1.863 | 1.859 | 9.996 | 9.482 | 0.146 | 0.131 |
| Design 5 | 1.852 | 1.841 | 8.434 | 7.991 | 1.413 | 1.377 |
| Design 6 | 1.859 | 1.856 | 9.617 | 9.492 | 0.149 | 0.158 |
| Design 7 | 1.838 | 1.839 | 7.967 | 8.025 | 1.213 | 1.357 |
| Design 8 | 1.848 | 1.845 | 7.846 | 7.981 | 1.283 | 1.403 |
| Design 9 | 1.830 | 1.825 | 8.088 | 8.017 | 1.123 | 1.264 |
| Design 10 | 1.855 | 1.842 | 8.534 | 8.014 | 1.240 | 1.382 |

## 5. Optimal Result and Experimental Verification

As presented in Table 3, the ratio of stator yoke and teeth width for different designs are set between 0.89 and 0.92. Considering that the area of stator slots will be reduced if a large value of $\lambda_{hcs}$ is selected. Thus, design 4 is selected as the final optimal solution, and $\lambda_{hcs}$ is finally determined as 0.89 due to the manufacturing requirement. The initial and optimal values of the optimization parameters are tabulated in Table 5. It can be found that, compared with the initial design, the average toque has been improved while the toque ripple and the difference between the adjacent phase currents have been reduced after optimization.

**Table 5.** Initial and optimal parameters of 12/10 SRM.

| Parameters | Initial | Optimal |
|---|---|---|
| $\lambda_{hcs}$ | 0.75 | 0.89 |
| $\alpha_{Aon}/°$ | 13 | 13 |
| $\alpha_{Bon}/°$ | 19 | 18.5 |
| $P_{dA}/°$ | 16 | 15.8 |
| $P_{dB}/°$ | 16 | 15.3 |
| $T_{avg}/Nm$ | 1.79 | 1.86 |
| $T_{rip}/\%$ | 11.1 | 9.3 |
| $I^*/\%$ | 8.5 | 0.5 |

After the determination of the optimal design, the control angles of the SRM are divided into two groups and an asymmetric control is utilized. For the asymmetric control, it means that the windings in phases A, C, and E are conducted in advance for 5° (corresponding to $\alpha_{Aon} = 13°$ in Table 5) and keep conducting for 15.8°, while those in phases B, D and F phases are conducted in advance for 5.5° (corresponding to $\alpha_{Bon} = 18.5°$ in Table 5) and keep conducting for 15.3°.

When the ratio of stator yoke and teeth width is defined as 0.89, the current and output torque under the proposed asymmetrical control ($\alpha_{Aon} = 13°$, $\alpha_{Bon} = 18.5°$) and the traditional symmetric control ($\alpha_{Aon} = 13°$, $\alpha_{Bon} = 19°$) methods are presented in Figure 8. In Figure 8, $I_{B1}$, $I_{C1,}$ and $T_1$ are the currents of phases B and C, and torque under the proposed asymmetric control, respectively, and $I_{B2}$, $I_{C2,}$ and $T_2$ are the currents of phases B and C, and torque under the traditional symmetric control, respectively. The difference is that, compared with the traditional symmetric control, the turn-on angle of phases B, D, and E of the proposed asymmetrical control is advanced by 0.5°. Thus, the current values in these phases can be enlarged, and the torque has been improved. It can be verified in Figure 8, obtained by FEM. Compared with the traditional symmetric control, the current of phase B and the minimum value of output torque can be improved by the proposed asymmetric control.

To validate the proposed method, the six-phase 12/10 SRM experimental platform based on dSPACE is constructed, as presented in Figure 9a. The rotational speed of

the SRM is defined as 1000 r/min due to the accuracy of the torque and speed sensor. Figure 9b,c present the torque and phase currents based on symmetrical control and asymmetric control, respectively. Channels 1, 2, and 3 are output torque and currents of phases B and C, respectively. As presented in Figure 9b,c, the peak value of torque for symmetrical control and asymmetric control are similar, and the valley value of torque can be enhanced through asymmetric control. Moreover, the phase current is more symmetrical for asymmetric control.

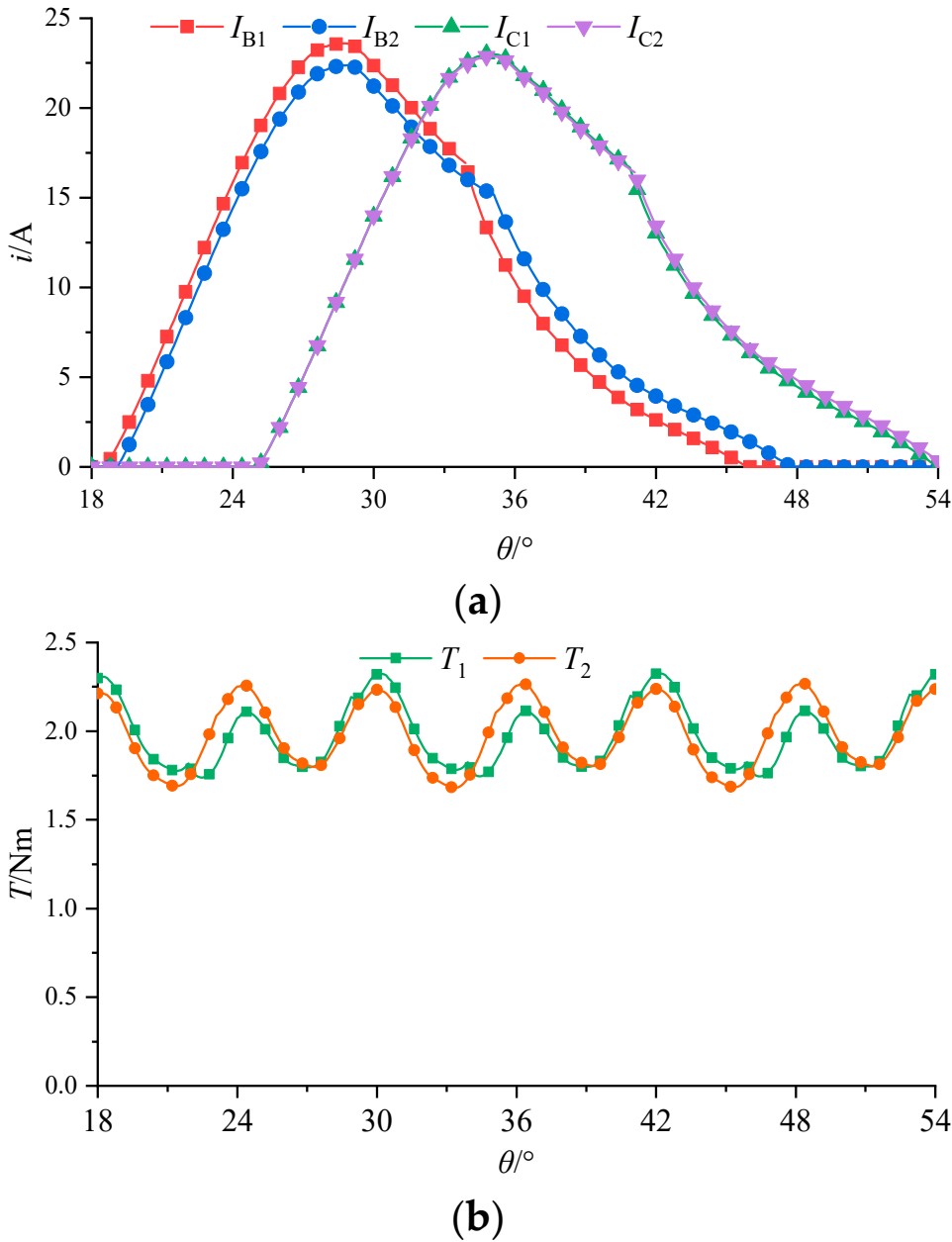

(a)

(b)

**Figure 8.** Comparison of (**a**) phase currents and (**b**) torque between the proposed asymmetric and the traditional symmetric control methods, where subscripts 1 and 2 represent the proposed asymmetric and the traditional symmetric methods, respectively.

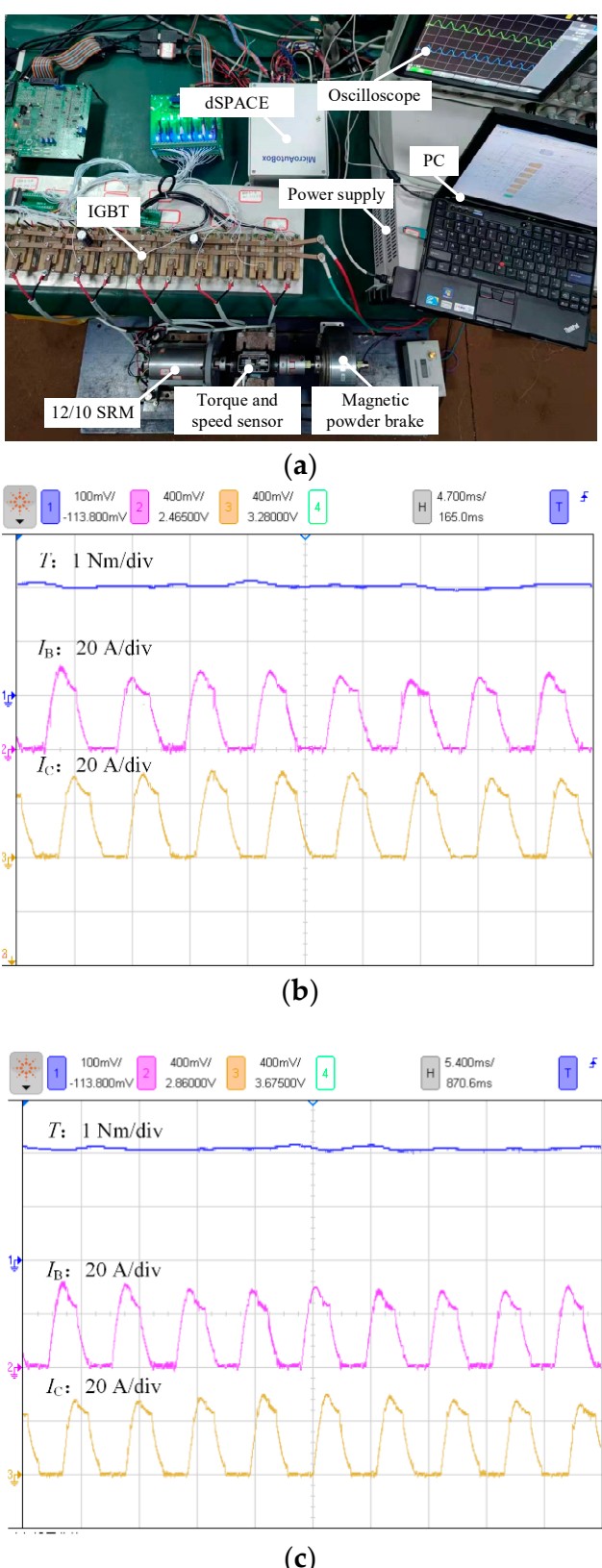

**Figure 9.** Experimental platform and results. (**a**) Test platform, (**b**) traditional symmetrical control, and (**c**) proposed asymmetric control.

## 6. Conclusions

In this paper, a multiobjective design optimization method incorporated with an asymmetric control was presented for a six-phase SRM. The asymmetric current phenomenon of the investigated SRM was introduced and further explained. It has been found that the stator yoke thickness, the turn-on angle, and the turn-off angle exhibit a great influence on the asymmetric phase currents. Three optimization objectives, i.e., the output torque, torque ripple, and normalization value of phase current symbolizing the difference between the adjacent phase currents, were selected. The Kriging model was established based on a certain number of finite-element samples, which can greatly reduce the computational time compared with directly using FEMs. The NSGA II was employed as an optimization algorithm to find the Pareto front. The final optimal design solution was selected from the Pareto front and analyzed. Finally, a prototype of the 12/10 SRM was manufactured, and the experimental platform was constructed to validate the proposed analysis and optimization method. The results show that the proposed asymmetric control method can improve the symmetrical characteristic of phase currents and reduce the torque ripple.

**Author Contributions:** Conceptualization, W.Q.; formal analysis, W.Q. and S.H.; investigation, W.Q. and S.H.; writing—original draft preparation, W.Q.; writing—review and editing, K.D. and X.S.; visualization, K.D.; supervision, X.S. All authors have read and agreed to the published version of the manuscript.

**Funding:** This research received no external funding.

**Institutional Review Board Statement:** Not applicable.

**Informed Consent Statement:** Not applicable.

**Data Availability Statement:** Not applicable.

**Conflicts of Interest:** The authors declare no conflict of interest.

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
