# Peer review of "Optimization Design and Control of Six-Phase Switched Reluctance Motor with Decoupling Winding Connections"

_applsci, doi:10.3390/app12178801_

Round 1
Reviewer 1 Report (New Reviewer)
The overall organization and the presentation of the manuscript's contents are worth appreciating. Authors have presented an approach to address the issue of Torque Ripple due to inconsistent phase current. Following are the comments.
1. Table 2 specifies the optimization parameters and their ranges. The revised manuscript should include a discussion on how these values/ranges are defined for optimization.
2. Section 3.2 discusses the basics of the kriging model. However, the discussion on how this kriging model is applied for 12/10 SRM is not found in the manuscript. Please include it in the revised manuscript.
3. Please include the discussion on the justification for the error between the kriging model results and the FEM model result in the revised manuscript.
4. The discussion on the stator geometry optimization is observed. However, there is no discussion on the optimization of control parameters. Could it be included in the revised manuscript?
Author Response
Please see the attachment.

Reviewer 2 Report (New Reviewer)
Line 10 (Abstract) - instead of "...switched reluctance motor (SRM) ..." it is proposed to replace by "... Switched Reluctance Motor (SRM) ..."
Lines 104-105 - it is recommended to replace “Figure1.” by “Figure 1.”
Line 138 (equation 1) – it is recommended to check the indices in the formula
Lines 291-292 - in the title of Figure 7 “(a)” and “(b)” should be bold
Line 304 - it is recommended to replace “Table.4.” by “Table 4.”
Line 306 - chapter 5 began with the Table 5 - it is recommended to start the chapter with the text (lines 308-312); the proposal is given in the attached file)
Line 306 - it is recommended to replace “Table.5.” by “Table 5.”
Line 324 - it is recommended to replace “Figure8.” by “Figure 8.”
Line 345 - descriptions in Figure 9a are illegible (they go beyond the white text boxes)
Lines 351-352 - it is recommended to replace “Figure9.” by “Figure 9.”
All suggestions are highlighted in the attached file.

Author Response
Please see the attachment.

Reviewer 3 Report (New Reviewer)
Line 13 and 250. Computational costs is claimed to have been reduced, but it is unclear by how much. What was the calculation time and memory size? It is not entirely clear what conclusions follow from Figure 6. Which variant of the finite element method was used? The aspects related to computational costs should be covered in more detail in the text of the article and in the conclusion.
Author Response
Please see the attachment.

Reviewer 4 Report (New Reviewer)
Dear Authors, please have a look to the requests in the attached file

Round 2
Reviewer 3 Report (New Reviewer)
Dear authors and editor!
I consider my previous comments taken into account.
However, why is FEMs plural in the added text?
Author Response
Please see the attachment.

This manuscript is a resubmission of an earlier submission. The following is a list of the peer review reports and author responses from that submission.
Round 1
Reviewer 1 Report
GENERAL COMMENTS
The paper deals with the design of six-phase switched reluctance motor (SRM). The authors applied the NSGA-II algorithm to optimize the design of the machine and also the control algorithm, as the optimized parameters are dimensions and phase turn-on angles. To reduce the computational burden , a Kriging-based surrogate model was introdced. Due to the relatively simple structure and low cost, SRMs are gaining interest as an alternative to machines with permanent magnets. Numerous papers were devoted to their design, esepcially to minimize their inherent drawbacks, such as torque ripple and acoustic noise. The present paper still makes a contribution to the current state-of-the-art, and its publications is justified. However, a major revision is necessary, to improve the quality of presentation.
The references cited are appropriate.
I advise English language revision, by a native speaker if possible. There are instances of unclear phrases, instances of inappropriate style and apparent calques from the Authors' first language. Some of them are listed in the specific comments.
The section 3.2. (Kriging model) should be thoroughly revised and rearranged. In the current form it is difficult to follow. Equation (10) is the expression of response of the Kriging model. In the next a correlation matrix and a correlation function (which should be denoted as r, not R, I think) are mentioned, but they are not in (10). A this point, f(x), β and z(x) should be explained, and the the other symbols. Also, what approximation model f(x) was used and how were the regression coefficients β estimated? What is the significance of σ2?
Figure 6. presents the comparison between the Kriging model and FEM. It is not clear what are the designs numbered on the x-axis. The optimized designs are introduced in the section 4 and only ten of them is mentioned there, not 15. Is the initial, non-optimum design one of them?
In the section 4, I strongly advice including the direct comparison of results for the initial and optimum designs, as it will make the presented case more convincing.
SPECIFIC COMMENTS
Page 1, line 6: an example of inappropriate style - the sentence is starting with "and". It can be dropped. Same for page 12, line 364.
Page 2, line 73: the subsection title should be capitalized.
Pages 2 and 3, lines 90-94: this is a very long and thus confusing sentence. It should be redrafted by breaking it into shorter phrases.
Subfigures in Fig. 1. lack the captions (a) and (b).
Page 4, line 133: beginning from this place, B [T] is in the paper called "magnetic induction intensity", which is confusing. It makes it look like a mix-up of B ("magnetic flux density", or "magnetic induction") and H ("magnetic field intensity"). I recommend to use "magnetic flux density" uniformly in the paper. The same applies for page 4, line 148, page 4, line 155 and page 5, line 162.
Page 4, line 139: what is "Comprehensive formulas (1) and (2) can be obtained" supposed to mean? I understand that (3) was derived from (1) and (2)
Page 4, line 146: should it be" the phase current of phase A can be expressed as"?
Page 5, line 172: the sentence "Therefore, the six-phase SRM finally selects the turn-on angle (...)" is baffling. I do not think it is the SRM that selects the turn-on angle.
Page 5, line 186, and page 6, line 188: this coefficient is commonly referred to as "slot fill factor".
As I have mentioned, the section 3.2. needs to be rearranged. The symbols should be introduced in the logical order. For example αk is first explained and then introduced as a part of r. Disorder makes this section difficult to follow.
Page 8, line 265: should it be "(...) which has been widely employed (...)"?
Table 4: please make it clear which results are from the Kriging model and from FEM. There is an index _s, but it is not explained.
Reviewer 2 Report
It has been a study that will make significant contributions to the literature. However, it is necessary to compare the results with similar studies in the literature. Self-citations by authors are too many. This should be reduced and replaced by similar studies in the literature.
Reviewer 3 Report
Ref.: Ms. No. applsci-1822131-peer-review-v1
Optimization Design and Control of Six-Phase Switched Reluctance Motor with Decoupling Winding Connections.
In this work, author shares an optimization design and control of six-phase switched reluctance motor with decoupling winding connections. The topic seems to be interesting. However, the quality of proposed work is not good according to the following comments:
1. In introduction section authors do not explain the significance of proposed method.
2. There is a major need to revise the article, because there are several grammatical mistakes in native English writing.
3. Overall presentation of paper is not attractive.
4. The advantages and disadvantages section is not included.
5. Novelty of new concept is not enough for publication.
***
Round 2
Reviewer 1 Report
The authors addressed all the concerns regarding the paper and made the necessary corrections in the revised version.
There is one thing left - page 13, line 457: "which has been wildly utilized". I had an issue with the word "wildly", which makes some sense in the context, but is quite colloquial and inappropriate in a scientific paper. I suppose it was meant to be "widely", understood as "extensively". I recommend correcting it in the final version of the paper.
Reviewer 3 Report
applsci-1822131-peer-review-v2
Title: Optimization Design and Control of Six-Phase Switched Reluctance Motor
with Decoupling Winding Connections.
The authors claims that this paper presents a new a design optimization method to reduce the current asymmetric and consequent torque ripple for a six-phase switched reluctance motor. However, the paper lacks a solid reasoning and discussion with respect to the presented claims and findings. Further, too many details are unclear and there are too many grammatically incorrect sentences. Moreover, the paper has been very poorly organized and presented. It is unconvincing to me the innovativeness and quality of the paper is sufficiently significant to considered to be published in this journal. The abstract of the paper reads poor. It should tell broad scope of the subject of the paper, gap between theory and practice and then what the paper does about it, some key results obtained.